# The Effect of Treatment-Induced Viral Eradication on Cytokine and Growth Factor Expression in Chronic Hepatitis C

**DOI:** 10.3390/v14081613

**Published:** 2022-07-24

**Authors:** Leona Radmanić, Kristian Bodulić, Petra Šimičić, Adriana Vince, Snježana Židovec Lepej

**Affiliations:** 1Department of Immunological and Molecular Diagnostics, University Hospital for Infectious Diseases “Dr. Fran Mihaljević”, 10000 Zagreb, Croatia; lradmanic@bfm.hr (L.R.); psimicic@bfm.hr (P.Š.); 2Research Department, University Hospital for Infectious Diseases “Dr. Fran Mihaljević”, 10000 Zagreb, Croatia; kbodulic@bfm.hr; 3Department of Viral Hepatitis, University Hospital for Infectious Diseases “Dr. Fran Mihaljević”, 10000 Zagreb, Croatia; avince@bfm.hr; 4School of Medicine, University of Zagreb, 10000 Zagreb, Croatia

**Keywords:** hepatitis C virus, chronic hepatitis C, fibrosis, cytokines, growth factors

## Abstract

In this study, we evaluated the effect of hepatitis C virus eradication using direct-acting antivirals (DAA) on the serum cytokine and growth factor profiles of chronic hepatitis C patients (CHC). Serum concentrations of 12 cytokines and 13 growth factors were measured in 56 patients with CHC before, during the DAA treatment and after sustained virological response using bead-based flow cytometry. Cytokine and growth factor levels were also measured in 15 healthy individuals. The majority of the selected cytokines and growth factors exhibited similar concentrations before, during and after successful DAA treatment, the exceptions being IL-10, EGF, HGF and VEGF. Significantly lower concentrations of IL-10, IL-13, IL-4, IL-4, IL-9, TNF- α and higher levels of Ang-2, HGF and SCF were observed in patients with CHC before and after DAA treatment compared with healthy individuals. Patients with severe fibrosis stages exhibited higher levels of Ang-2 and lower levels of EGF, PDGF-AA and VEGF. Furthermore, IL-4, IL-5 and SCF were characterized as potential biomarkers of DAA treatment using random forest. Additionally, logistic regression characterized EGF as a potential biomarker of severe CHC. Our results suggest inhibition of pro-inflammatory processes and promotion of liver regeneration in CHC patients during DAA treatment.

## 1. Introduction

Hepatitis C virus (HCV) is a hepatotropic RNA virus and a major cause of chronic hepatitis C (CHC) worldwide [1]. The importance of interactions between HCV virions and hepatocytes is significant for a better understanding of immunoreactions and immunopathogenesis in HCV infection [2]. Treatment of HCV infection is based on the use of direct-acting antivirals (DAA) that inhibit different phases of the HCV replication cycle and allow viral eradication in more than 95% of infected individuals [3]. In general, DAAs can be divided into three major classes based on their molecular targets: the nonstructural protein 3/4A (NS3/4A) protease inhibitors, NS5A inhibitors and NS5B polymerase inhibitors, the combination of which is used for CHC treatment [4]. Replication of HCV in hepatocytes and the inflammatory response of the immune system during chronic infection activate numerous mechanisms that lead to fibrosis, cirrhosis and ultimately hepatocellular carcinoma (HCC), as a complication of CHC [5].

Chronic HCV infection is characterized by direct modulation of signaling and metabolic pathways mediated by viral proteins, as well as the induction of antiviral immune responses which lead to chronic inflammation and the development of liver fibrogenesis [6]. Liver fibrosis is a complex pathological process that represents the accumulation of connective tissue in the liver in response to hepatocellular damage that occurs due to strong formation or insufficient degradation of the extracellular matrix (ECM). A major starting event in the development of fibrosis is the activation of liver stellate cells (HSC), which are the primary source of ECM. In response to cell injury, ECM activation mediates various biological response modifiers, including reactive species oxygen, lipid peroxides, inflammatory cytokines and growth factors [7,8]. Despite evidence of fibrosis reversibility after HCV eradication, exact mechanisms are currently unknown [8].

Cytokines are glycoproteins that mediate intercellular interactions and promote proliferation, differentiation, growth or apoptosis of target cells, regulate the host immune response and play an important role in the pathogenesis of various diseases [9]. Inflammatory responses caused by HCV infection can cause progressive liver disease. Some inflammatory cytokines may serve as biomarkers for monitoring disease progression and treatment outcome in patients with CHC [10]. The pathogenesis of HCV-infected patients is complicated and includes classical pathogen recognition, inflammatory activation, intrahepatic inflammatory cascade response, oxidative stress and endoplasmic reticulum stress. Persistent replication of HCV in hepatocytes leads to uncontrolled inflammation. Cytokines, as inflammatory agents, can cause inflammation in the liver leading to liver tissue damage and progression of liver disease [7]. The interaction of liver parenchymal, non-parenchymal cells and immune cells migrating to the liver during CHC is mediated by biological response modulators such as cytokines and growth factors. Both hepatic parenchymal and nonparenchymal cells are involved in the initiation and progression of liver fibrosis and cirrhosis [11].

Growth factors play an important role in the immunopathogenesis of liver fibrosis and show profibrotic and antifibrotic effects in patients with CHC. They can contribute to fibrogenesis and the development of cirrhosis. However, they can also play a part in liver regeneration and tissue repair [12]. Further research is essential to gain a better understanding of molecular mechanisms associated with the pathogenesis of fibrosis and the effect of virus elimination using DAA in CHC patients.

This study aimed to evaluate the effect of HCV eradication using antiviral drugs on the serum cytokine and growth factor profiles of CHC patients. This would allow for a better understanding of the physiology of immunoreactions and the mechanisms of fibrosis in CHC.

## 2. Materials and Methods

### 2.1. Patients

This study is a retrospective analysis of cytokine and growth factor levels in patients with CHC who received clinical care at the Croatian Reference Center for Viral Hepatitis, University Hospital for Infectious Diseases (UHID) in Zagreb, Croatia. The study enrolled 56 CHC patients (≥18 years old, treated with DAA) and an age and sex-matched control group of 15 healthy individuals. All patients achieved sustained virological response (SVR) defined as undetectable HCV RNA at 12 weeks after therapy completion. Patients with known hematological, malignant (incl. hepatocellular carcinoma) or autoimmune diseases were excluded from the study. Ethics committees of the UHID approved the study (code 01-673-1-2021) and all subjects signed a consent form.

Liver fibrosis was analysed with noninvasive transient elastography (TE) using FibroScan devices (Echosens, France). Fibrosis stages were classified according to METAVIR score as: F0/F1 < 7.0 kPa, F2 7.0–9.5 kPa, F3 > 9.5 kPa and F4 > 12.5 kPa [13].

### 2.2. Serum Sample Collection and Virological Monitoring

Cytokine and growth factor concentrations in patients with CHC were determined at four different time points. The first time point represented serum samples obtained from patients with CHC before antiviral treatment. Serum samples from the same patients were also collected after four, eight and 12 weeks of treatment (second, third and fourth time point). The samples were stored at −80 °C to avoid repeated freeze and thaw cycles. Quantification of HCV RNA was performed with COBAS AmpliPrep/COBAS TaqMan HCV Test (Roche Diagnostics, Manheim, Germany) as recommended by the manufacturer.

### 2.3. Bead-Based Cytometry for Cytokine and Growth Factors Levels Analysis

In this study, we measured the serum levels of 12 cytokines (interleukin 5 (IL-5), IL-13, IL-2, IL-6, IL-9, IL-10, interferon-gamma (IFN-γ), tumor necrosis factor-alpha (TNF-α), IL-17A, IL-17F, IL-4 and IL-22) and 13 growth factors (angiopoietin-2 (Ang-2), epidermal growth factor (EGF), erythropoietin (EPO), fibroblast growth factor-basic (FGF-basic), granulocyte colony-stimulating factor (G-CSF), granulocyte-macrophage colony-stimulating factor (GM-CSF), hepatocyte growth factor (HGF), macrophage colony-stimulating factor (M-CSF), platelet-derived growth factor (PDGF-AA), PDGF-BB, stem cell factor (SCF), transforming growth factor-alpha (TGF-α) and vascular endothelial growth factor (VEGF). Human Th Cytokine Panel (12 plex) with Filter Plate (Biolegend, San Diego, CA, USA) was used to determine cytokine concentrations and Human Growth Factor Panel (13 plex) with Filter Rate (Biolegend, San Diego, CA, USA) was used to determine growth factor levels according to manufacturer instructions. These beads were mixed with 25 mL of serum samples. After incubation, the biotin-conjugated second antibody was added and analyzed cytokines and growth factors were detected using streptavidin-phycoerythrin which binds to the biotin conjugate and emits a fluorescent signal. The detection process was carried out on flow cytometer BD FACSCanto II (Beckton Dickinson, Franklin Lakes, NJ, USA). Finally, the concentrations of the analysed cytokines and growth factors were determined using LEGENDplex (version 8.0., Biolegend, San Diego, CA, USA) software.

### 2.4. Statistical Analysis

Data visualization and analysis were done in R (version 4.1.0., R Core Team 2022, Vienna, Austria) [14]. Cytokine and growth factor levels at different time points were compared using the Friedman test and the Wilcoxon signed-rank test. The Mann–Whitney U test was used when comparing cytokine and growth factor levels between patients and healthy individuals. The correlation between cytokine and growth factor levels was evaluated with Spearman’s correlation coefficient and correlation test. Using cytokine and growth factor levels, random forest models were utilized to classify subjects into patient and healthy control groups. The number of trees was set to 10,000 and five variables were considered at every split point. Models’ performance was evaluated with out-of-bag error estimates and predictor importance was estimated by calculating the average Gini index reduction for every predictor. Cytokine and growth factor levels in patients with different liver fibrosis stages were compared with the Mann-Whitney U test. Patients were classified by liver fibrosis levels using binary logistic regression, where predictors were chosen by best subset selection. The model was evaluated with five-fold cross-validation and receiver operating characteristic (ROC) curve. Predictor cut-off points were determined using single dichotomization with the maximal Youden index criterium. All statistical tests were two-tailed with a significance level of 95%. *p*-values were corrected for multiple testing with the Hommel method.

## 3. Results

### 3.1. Patients’ Characteristics

A total of 56 CHC patients and 15 age and sex-matched healthy controls were included in this study (Table 1). The mean age of patients was 56.7 years (range 29–75 years) and 48.2% of them were male. The mean viremia of patients before starting antiviral therapy was 923,808 HCV RNA copies/mL (range 2463–9,550,925 HCV RNA copies/mL). All participants achieved SVR after DAA treatment. A total of 73.2% of patients were infected with HCV genotype 1, 23.2% of patients with HCV genotype 3, 5.4% of patients with HCV genotype 2 and 1.8% of patients with HCV genotype 4. When considering liver fibrosis before treatment, 17.9% of patients exhibited mild fibrosis (F1), 25.0% of patients exhibited medium fibrosis (F2), 8.9% of patients showed severe fibrosis (F3) and 48.2% of patients had liver cirrhosis (F4). After DAA treatment, 71.4% of patients showed an unchanged METAVIR fibrosis score, 23.2% of patients exhibited a decrease in a fibrosis score of one, 8.9% of patients exhibited a decrease in a fibrosis score of two and 1.8% of patients showed a decrease in a fibrosis score of three.

### 3.2. Cytokine and Growth Factor Levels in CHC Patients’ Sera before, during and after DAA Treatment

In order to gain a better understanding of cytokine and growth factor dynamics in the course of CHC, we measured the levels of 12 cytokines (IFN-γ, IL-10, IL-13, IL-17A, IL-17F, IL-2, IL-22, IL-4, IL-5, IL-6, IL-9 and TNF-α) and 13 growth factors (Ang-2, EGF, EPO, FGF-basic, G-CSF, GM-CSF, HGF, M-CSF, PDGF-AA, PDGF-BB, SCF, TGF-α and VEGF) in sera taken at four different time points (before treatment, after four and eight weeks of DAA therapy and 12 weeks after treatment, e.g., SVR12). Cytokine and growth factor level comparison at the four selected time points is shown in Figure 1. The majority of the measured cytokines exhibited similar concentrations before treatment, after four and eight weeks of therapy and 12 weeks after DAA treatment, except IL-10, which showed a steady decrease at selected time points (medians 1.2, 1.1, 1.1 and 0.9 pg/mL, *p* = 0.038). Concentrations of the analysed cytokines were relatively low.

In addition, most of the analysed growth factors displayed similar levels before treatment, after four and eight weeks of therapy and 12 weeks after DAA treatment, except for EGF, HGF and VEGF. Out of these growth factors, EGF exhibited the highest relative increase across the four selected measurement time points (medians 18.2, 71.3, 54.7 and 73.4 pg/mL, *p* < 0.001), followed by HGF (medians 568.3, 698.5, 775.4 and 848.7 pg/mL, *p* = 0.007) and finally VEGF (medians 139.7, 151.1, 163.1 and 156.3 pg/mL, *p* = 0.040). Notably, EGF, HGF and VEGF levels were significantly higher four weeks after antiviral therapy onset when compared to the respective levels before starting antiviral therapy (*p* < 0.001, *p* = 0.005 and *p* = 0.041, respectively) and remained similar in later time points (*p* > 0.05).

We also examined the relationship between cytokines and growth factors of interest. The correlation matrix of the measured cytokines and growth factors in patients before starting antiviral treatment is shown in Appendix A. In particular, we found a moderate positive correlation between IFN-γ and FGF-basic (r = 0.57, *p* < 0.001), G-CSF (r = 0.46, *p* < 0.001), GM-CSF (r = 0.50, *p* < 0.001). We also found a moderate positive correlation between IL-17A and FGF-basic (r = 0.47, *p* < 0.001). When considering viremia, we found a weak positive correlation between HCV RNA serum levels and PDGF-AA (r = 0.30, *p* = 0.002), PDGF-BB (r = 0.29, *p* = 0.002) and EGF (r = 0.27, *p* = 0.003). We did not find a significant correlation between viremia and cytokine and growth factor levels at other analysed time points (*p* > 0.05).

### 3.3. Comparison of Cytokine and Growth Factor Levels in CHC Patients and Healthy Individuals

We also compared cytokine and growth factor concentrations in CHC patients before treatment and after SVR with healthy controls (Figure 2). Cytokines which were significantly lower in CHC patients before treatment onset compared to healthy individuals were: IL-10 (medians 1.2 and 3.0 pg/mL, *p* = 0.007), IL-13 (medians 8.2 and 14.9 pg/mL, *p* < 0.001), IL-4 (medians 1.7 and 2.0 pg/mL, *p* < 0.001), IL-5 (medians 4.5 and 12.8 pg/mL, *p* = 0.041), IL-9 (medians 3.6 and 6.0 pg/mL, *p* = 0.005) and TNF-α (medians 4.5 and 6.0 pg/mL, *p* = 0.009). Furthermore, cytokines which exhibited significantly lower levels in patients after reaching SVR when compared to the control group were IL-10 (medians 0.9 and 3.0 pg/mL, *p* < 0.001), IL-13 (medians 8.3 and 14.9 pg/mL, *p* < 0.001), IL-4 (medians, *p* = 1.7 and 2.0 pg/mL, *p* < 0.001), IL-5 (medians 4.5 and 12.8 pg/mL, *p* = 0.048), IL-9 (medians 3.6 and 6.0 pg/mL, *p* < 0.001) and TNF-α (medians 4.5 and 6.0 pg/mL, *p* = 0.032).

Growth factors which displayed significantly higher levels in patients before starting antiviral treatment when compared to the control group were Ang-2 (medians 2454.0 and 1206.6 pg/mL, *p* = 0.018), HGF (medians 568.3 and 374.7 pg/mL, *p* = 0.041) and SCF (medians 96.4 and 28.6 pg/mL, *p* < 0.001). Likewise, growth factors which showed significantly higher levels in patients after reaching SVR when compared to the healthy individuals were Ang-2 (medians 2180.4 and 1206.6 pg/mL, *p* = 0.027), EGF (medians 73.4 and 30.8 pg/mL, *p* = 0.037), HGF (medians 848.7 and 374.7 pg/mL, *p* = 0.006) and SCF (medians 86.5 and 28.6 pg/mL, *p* < 0.001).

In order to assess the relative importance of cytokines and growth factors in CHC pathology, we used the random forest algorithm to classify the participants based on cytokine and growth factor levels measured before antiviral treatment in the patient and healthy control groups. This model achieved an out-of-bag error of 4.2% (98.2% sensitivity and 86.7% specificity). We also used the random forest algorithm to classify patients based on cytokine and growth factor levels measured after reaching SVR in the patient and healthy control groups. This model achieved an out-of-bag error of 5.6% (96.4% sensitivity and 86.7% specificity). The relative importance of cytokines and growth factors in classifying patients and healthy individuals is shown in Figure 3. Notably, IL-4, SCF and IL-5 were the most important predictors for separating patients before starting antiviral therapy from the control group, as well as patients after reaching SVR from the control group.

### 3.4. Cytokine and Growth Factor Levels in CHC Patients with Different Fibrosis Stages

In order to inspect the relationship of cytokines and growth factors with CHC severity, we compared the levels of analysed cytokines and growth factors in participants with different liver fibrosis levels before starting DAA treatment. In doing so, we stratified the analysed patients into two groups, where the first group consisted of patients with fibrosis stages F1 and F2, whereas the second group consisted of patients with fibrosis stages F3 and F4. The results of these comparisons are shown in Figure 4. None of the measured cytokines were significantly different in patients with higher fibrosis levels before starting DAA treatment. However, when evaluating the levels of analysed growth factors, we found that Ang-2 showed significantly higher levels in patients with higher fibrosis levels when compared to patients with lower fibrosis levels (medians 2646.2 and 1769.8 pg/mL, *p* = 0.045). Furthermore, several growth factors showed significantly lower concentrations in patients with higher fibrosis levels. These growth factors included EGF (medians 7.2 and 56.7 pg/mL, *p* < 0.001), G-CSF (medians 51.5 and 61.1 pg/mL, *p* = 0.044), PDGF-AA (medians 45,585.6 and 61,272.0 pg/mL, *p* = 0.032) and VEGF (medians 100.2 and 163.0 pg/mL, *p* = 0.038). Furthermore, patients with a reduction in fibrosis levels after DAA treatment exhibited lower levels of Ang-2 before DAA treatment when compared to patients with unchanged fibrosis levels (medians 2182.5 and 3095.8 pg/mL, *p* = 0.031).

In order to evaluate the relationship of cytokine and growth factor levels with CHC severity in a multivariate environment, we used binary logistic regression to classify the patients into F1/F2 and F3/F4 groups defined by liver fibrosis levels before receiving DAA treatment. Predictors that best separated the groups were EGF, Ang-2 and patient’s sex. When doubling the value of EGF, the odds of a patient exhibiting F3 or F4 fibrosis stage before antiviral treatment decreased by 2.00 times (95% CI 1.43–3.14, *p* < 0.001). Furthermore, when doubling the value of Ang-2, the odds of a patient exhibiting F3 or F4 fibrosis stage before antiviral treatment increased by 4.19 times (95% CI 1.55–14.34, *p* < 0.001). Finally, male patients were 8.23 times more likely to exhibit F3 or F4 fibrosis stage than female patients (95% CI 1.76–36.88, *p* = 0.009). We tested the accuracy of this model with cross-validation, which resulted in an error rate of 17.8% (sensitivity 83.1%, specificity 80.0%). The model’s ROC curve exhibited an AUC of 0.85. We also evaluated the possible usage of EGF and Ang-2 as sole predictors of severe fibrosis. The model using EGF as a sole predictor of severe fibrosis achieved a sensitivity of 75,0% and a specificity of 87.5% when using cross-validation. Furthermore, the model using Ang-2 as a sole predictor of severe fibrosis had a sensitivity of 60.4% and a specificity of 80.7%. The models’ ROC curves exhibited an AUC of 0.84 and 0.67, respectively (Appendix A). The cut-off values for severe fibrosis were 26.7 pg/mL for EGF and 1972.6 pg/mL for Ang-2.

## 4. Discussion

In this study, we analyzed the dynamics of cytokines and growth factors of interest in CHC patients at four time points after DAA treatment. Interestingly, the levels of pro-inflammatory cytokines (IL-6 and TNF-α), Th1 cytokines (IL-2 and IFN-γ), Th2 cytokines (IL-4, IL-5 and IL-13), Th 9 cytokines (IL-9) and Th17 cytokines (IL-17A, IL-17F, and IL-22) were relatively low and did not significantly change in response to virus eradication by DAA. However, we showed a continuous decrease in serum IL-10 levels across the four disease stages. It is known that decreased IL-10 expression in the liver is associated with greater inflammatory response and infiltration of activated cells of the innate and specific immune system. These conditions can ultimately lead to fibrosis progression and cirrhosis [15]. However, the stable and relatively low levels of other analysed cytokines suggest that DAA treatment did not induce significant changes in the immune response of CHC patients, including a pro-inflammatory response that could promote tumorigenesis. This is especially important when considering the evidence for tumor recurrence in CHC patients that achieved SVR [16].

When comparing cytokine levels in CHC patients and healthy individuals, we found that CHC patients showed lower levels of proinflammatory cytokine TNF-α, anti-inflammatory cytokine IL-10, Th2 cytokines (IL-4, IL-5 and IL-13), Th22 cytokines (IL-22) and Th9 cytokines (IL-9) before starting DAA treatment and after achieving SVR. This finding partially goes in line with a study showing significantly reduced concentrations of IL-5, IL-9 and IL-10 in patients with CHC compared to healthy controls [17]. In addition, Riberio et al. suggested that lower levels of IL-5 could be a universal biomarker of DAA treatment and an important factor in the case of HCV re-infection [17]. Furthermore, Baskic et al. showed that lower but sustained IL-4 production refers to Th2 predominance in higher stages of fibrosis [18]. Considering that CHC is generally characterised by a strong proinflammatory response, these findings provide more evidence for the potential inhibition of inflammatory mechanisms during DAA treatment of CHC patients. Notably, Riberio et al. found elevated serum concentrations of IL-4, TNF-alpha, FGF, PDGF and GM-CSF in patients with CHC compared to healthy controls [17]. This discrepancy could potentially be attributed to higher age and higher fibrosis levels of patients analysed in the Brazilian study. All in all, our results suggest the inability of DAA treatment to completely restore altered cytokine levels in CHC, as already suggested by other studies [19].

When analysing growth factor levels in different stages, CHC, EGF, HGF and VEGF exhibited a significant increase four weeks after antiviral treatment when compared to their respective levels before DAA treatment. Cienfuegos et al. demonstrated that EGF is effective in liver regeneration [20]. Considering that the concentration of EGF in our study was the highest after the treatment of CHC infection, our findings could indicate liver tissue repair after treatment with DAA. Likewise, HGF is a pleiotropic cytokine produced by hepatic stellate cells (HSC) and involved in liver regeneration during acute and chronic damage and fibrosis [21]. Serum HGF concentrations depend on the degree of liver damage and indicate the degree of hepatocellular dysfunction. Additionally, Marín-Serrano et al., found that HGF concentration was an independent factor associated with the degree of fibrosis in CHC [22]. Our study showed that the concentration of HGF is the highest after achieving SVR, indicating the possibility that increased concentrations of HGF after treatment are a consequence of liver damage and affect the regeneration of liver tissue. Growth factor VEGF also plays an important role in the immunopathogenesis of liver fibrosis, especially in the wound healing process that is common in chronic liver disease [23]. Although VEGF levels mildly increased during DAA treatment, they were not significantly different from the VEGF levels of healthy individuals. Taken together, the recorded increase of EGF and HGF levels suggests a considerable degree of liver tissue remodeling and regeneration in CHC patients during DAA treatment.

When considering growth factor level differences between CHC patients and the control group, we found that serum levels of Ang-2, HGF and SCF were higher both before starting antiviral treatment and after SVR when compared to healthy individuals. Similarly, we found that EGF levels were significantly higher only after reaching SVR when compared to healthy controls. These findings add another piece of evidence to the hypothesis of EGF and HGF being important contributors to liver regeneration in CHC patients during DAA treatment. Although Ang-2 and SCF levels did not significantly change in response to DAA therapy, the levels of these growth factors significantly increased compared to those of healthy individuals. These findings suggest that Ang-2 and SCF contribute to liver repair in CHC, regardless of DAA treatment.

By analyzing the correlation matrix of cytokine and growth factor expression before initiating treatment, a significant correlation was demonstrated between the expression of cytokine IFN-γ and growth factors FGF-basic, G-CSF, GM-CSF as well as between cytokine IL-17A and growth factor FGF-basic. This could be explained by a strong coherence between pro-inflammatory and pro-fibrotic factors before DAA treatment. However, this hypothesis should be further evaluated.

Using the random forest algorithm, we classified patients and healthy individuals by their cytokine and growth factor levels with high accuracy. Moreover, models classifying patients before starting treatment and patients after achieving SVR exhibited remarkably high levels of sensitivity and specificity. The most important biological modulators in classifying patients and healthy individuals in both models were IL-4, SCF and IL-5, underlining the already discussed importance of these cytokines and growth factors in CHC pathology. This finding points towards a possible usage of IL-4, SCF and IL-5 as biomarkers for DAA treatment of CHC.

Finally, we sought to assess the relationship between analysed biological modulators and CHC severity. When comparing cytokine levels before antiviral treatment in patients with varying stages of fibrosis, we found no significant differences. This finding goes in hand with the already discussed low pro-inflammatory cytokine expression and suggests a mechanism of fibrosis formation in CHC independent of inflammation, which has already been shown in experimental models [24]. Considering that various studies report a significant correlation between inflammation and fibrosis in CHC, this represents a novel finding [25]. When analysing growth factor levels, we found that patients with higher fibrosis levels exhibited higher Ang-2 levels and lower EGF, G-CSF, PDGF-AA and VEGF levels. It is known that Ang-2 plays a crucial role in angiogenesis and is, therefore, an important contributor to HCC progression. Hernández-Bartolomé et al. showed that the ratio of Ang-2 and Ang-1 concentrations is significantly correlated with the degree of fibrosis in patients with CHC [26]. This finding goes in line with our results and shows that Ang-2 could be important in severe HCC pathogenesis [26]. To our knowledge, the finding of lower EGF levels in severe fibrosis is novel and has not been discussed in the literature. Even though the research on G-CSF in the context of liver fibrosis has been scarce, Meng et al., showed that G-CSF is one of the important factors in remodeling liver tissue with EPO, SCF and GM-CSF [27]. Furthermore, it has been suggested that the administration of G-CSF reduces the mortality of chronic liver failure. In the context of EGF and G-CSF, our results could suggest a lower degree of tissue remodeling in patients with higher fibrosis levels. Finally, it has been shown that increased PDGF expression is important in the development of fibrosis [28]. Our results contradict this hypothesis and imply differing roles of PDGF-AA in liver fibrosis.

The binary logistic regression model was used to classify patients by liver fibrosis levels before receiving antiviral treatment (F1/F2 vs. F3/F4). This analysis revealed an independent association of EGF, Ang-2 and male sex with the degree of fibrosis. Therefore, the logistic regression confirms the importance of EGF and Ang-2 found in the bivariate analysis. Finally, male patients were more likely to exhibit F3 or F4 fibrosis stage than female patients. If not due to higher alcohol consumption and poor diet in the studied male patient group, this new finding should be evaluated in future research. Notably, our regression model with EGF as a sole predictor of severe fibrosis achieved relatively high accuracy in classifying patients by their fibrosis levels, which points towards a potential usage of EGF as a biomarker of severe fibrosis in CHC.

The main limitation of this study is the lack of cytokine and growth factor level monitoring after SVR. This could allow for a more detailed immune response and liver remodeling assessment. Another shortfall of this study is a small percentage of HCV genotypes 2 and 4 among analysed patients, which did not allow for the evaluation of the effect of HCV genotype on cytokine and growth factor levels. However, considering that the HCV genotype ratios of patients included in this study reflect the HCV genotype ratio in the population, we believe that this did not significantly affect the conclusions of our study.

## 5. Conclusions

Findings presented in this study suggest the potential inhibition of pro-inflammatory processes and significant promotion of liver remodeling and regeneration in CHC patients during DAA. Furthermore, IL-4, IL-5 and SCF were characterized as potential biomarkers of DAA treatment, whereas EGF was shown to be a potential biomarker of severe CHC. Additional analysis of modulators of biological responses important in CHC pathology will significantly contribute to the better elucidation of the physiology of immunoreactions in chronic HCV infection and mechanisms of fibrosis.

## Figures and Tables

**Figure 1 viruses-14-01613-f001:**
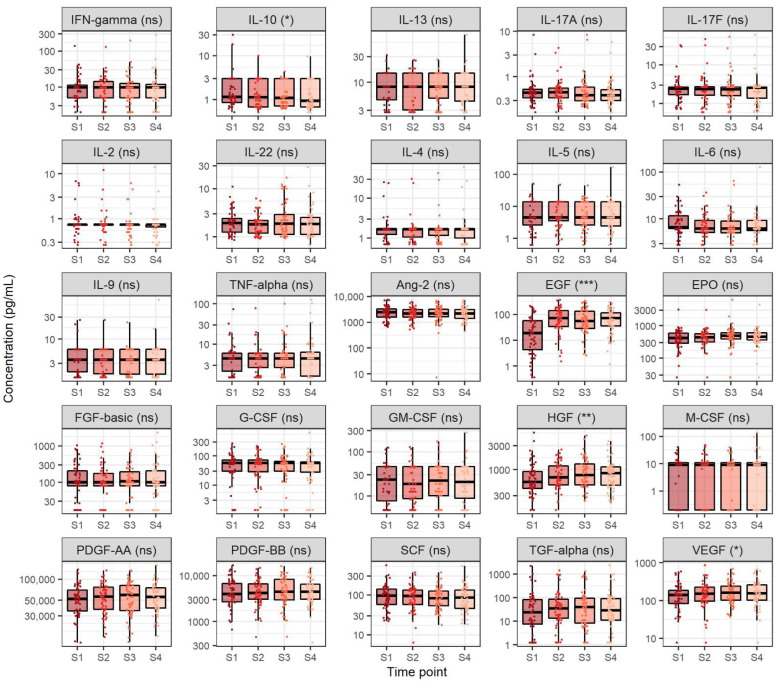
Comparison of cytokine and growth factor levels in chronic hepatitis C at four selected timepoints. The boxes show the median and interquartile range of the distribution, whereas the whiskers extend to the minimum and maximum nonoutlier values of the distribution. *Y*-axis is logarithmically scaled. S1: before antiviral treatment, S2: four weeks after treatment onset, S3: eight weeks after treatment onset, S4: sustained virological response, ns: non-significant, *: *p* < 0.05, **: *p* < 0.01, ***: *p* < 0.001 (Friedman’s test, *p*-values corrected with the Hommel method).

**Figure 2 viruses-14-01613-f002:**
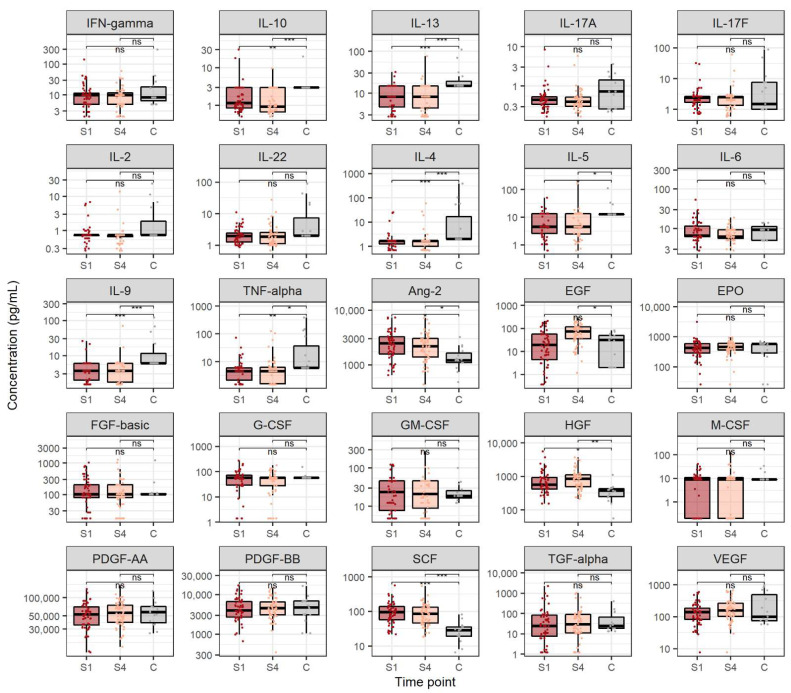
Cytokine and growth factor levels in chronic hepatitis C before antiviral treatment and after sustained virological response compared to healthy individuals. The boxes show the median and interquartile range of the distribution, whereas the whiskers extend to the minimum and maximum nonoutlier values of the distribution. *Y*-axis is logarithmically scaled. S1: before antiviral treatment, S4: sustained virological response, C: healthy control, ns: non-significant, *: *p* < 0.05, **: *p* < 0.01, ***: *p* < 0.001 (Mann–Whitney U test, *p*-values corrected with the Hommel method).

**Figure 3 viruses-14-01613-f003:**
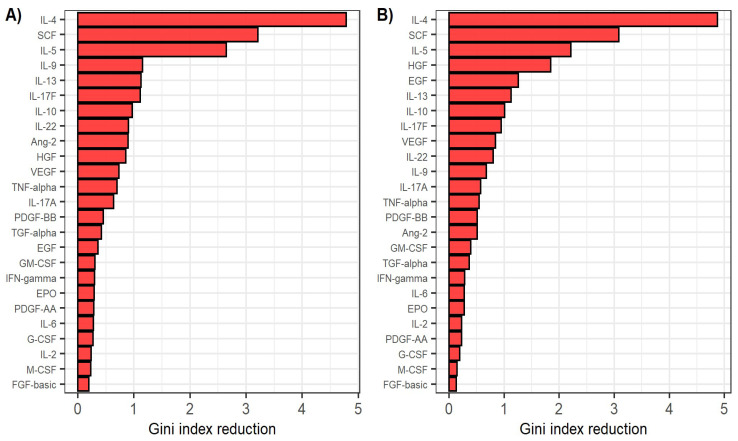
(**A**) Relative variable importance in classifying patients before antiviral treatment and healthy individuals by the random forest classifier. (**B**) Relative variable importance in classifying patients after reaching sustained virological response and healthy individuals by the random forest classifier. Variable importance is defined as the mean decrease in Gini coefficient when using the respective predictor in separating the groups.

**Figure 4 viruses-14-01613-f004:**
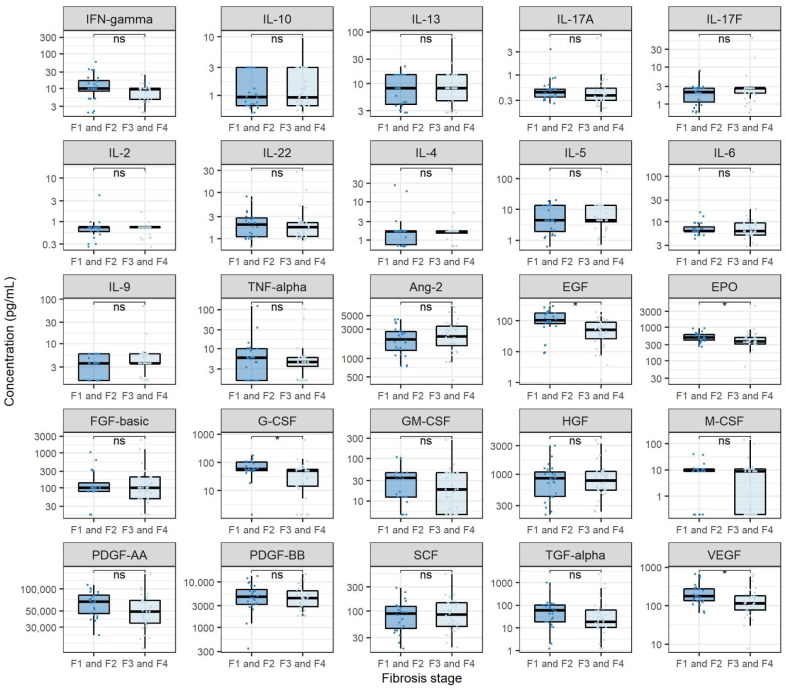
Comparison of cytokine and growth factor levels in chronic hepatitis C patients with different fibrosis stages. The boxes show the median and interquartile range of the distribution, whereas the whiskers extend to the minimum and maximum nonoutlier values of the distribution. *Y*-axis is logarithmically scaled. F1: mild fibrosis, F2: medium fibrosis, F3: severe fibrosis, F4: liver cirrhosis, ns: non-significant, *: *p* < 0.05 (Mann–Whitney U test, *p*-values corrected with the Hommel method).

**Table 1 viruses-14-01613-t001:** Patients’ demographic and clinical characteristics.

Characteristic	
Age, years, mean (range)	56.7 (29–75)
Male sex N (%)	27 (48.2%)
HCV viremia, copy/mL, mean (range)	923,808 (2463–9,550,925)
HCV genotype N (%)	
1	41 (73.2%)
3	13 (23.2%)
2	3 (5.4%)
4	1 (1.8%)
METAVIR fibrosis score N (%)	
F1	10 (17.9%)
F2	14 (25.0%)
F3	5 (8.9%)
F4	27 (48.2%)
METAVIR fibrosis score decrease after DAA treatment N (%)	
0 (no changes)	40 (71.4%)
−1	13 (23.2%)
−2	2 (3.6%)
−3	1 (1.8%)

HCV = hepatitis C virus, DAA = direct-acting antiviral.

## Data Availability

The data presented in this study are available in this published article and references.

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
