# Peer review of "The Effect of Treatment-Induced Viral Eradication on Cytokine and Growth Factor Expression in Chronic Hepatitis C"

_viruses, 2022, doi:10.3390/v14081613_

Round 1
Reviewer 1 Report
In this study, the authors used beads based FACS assay to determine the level of cytokines during DDA therapy in chronic HCV patients. The authors also correlate the level of cytokines to the fibrosis stage. The authors found that lower concentrations of IL-10, IL-13, IL-4, IL-4, IL-9, TNF- α and higher levels of Ang-2, HGF and SCF were observed in patients with CHC before and after DAA treatment compared with healthy individuals .
I have major comments
1- Why the authors used FACS not ELISA in this study? The ELISA is simple and most of studies use the ELISA assay. The authors need to explain this point in the introduction and discussion. Also, did the authors have any correlation between FACS assay and ELISA assays regarding the cytokines.
2- Please add a table include all demographic criteria of CHC patients and healthy control.
3- The authors mentioned in the methodology that they assessed HCV RNA by qPCR. Please correlate between HCV RNA and cytokines level at different time points
4- The authors should generate ROC curves to determine the cut off of cytokines that can affect the fibrosis stage, and please mention the sensitivity and specificity.
5- Are all patients have SVR during therapy? if resistance or relapse has developed, please verify and mention the level if assessed cytokines in these cases
6- Is there hepatocellular carcinoma developed in any of these patients? If yes, please verify this point and level of cytokines
7- Can we predict the treatment outcomes and fibrosis stage development by measuring the cytokines before, after therapy? The authors need to do more analytical and statistical work
Author Response
Reviewer 1:
Dear reviewer, we thank you for your comments, suggestions and the time taken to review our manuscript. Here are our remarks:
We used bead-based flow cytometry instead of ELISA because it enabled us to quantify 12 cytokines and 13 growth factors from a total of 50µL of sample (25µL for each panel). The amount of samples that we would require to quantify 25 biological response modifiers by ELISAs that tipically require 100-200µL of sample would not be possible due to the limited amount of samples available for clinical studies. The bead-based flow cytometry approach is a standard and frequently used method in immunological studies that involve cytokine/growth factor quantification in biological samples from patients instead of ELISA (the results are comparable). Since bead-beased flow cytometry it is a standard method that is used so often we feel that this issue is self-explanatory. Please referr to some of the recent publications from our group:
M, Haberle S, Židovec-Lepej S, Savić V, Kusulja M, Papić N, Višković K, Župetić I, Savini G, Barbić L, Tabain I, Kutleša M, Krajinović V, Potočnik-Hunjadi T, Dvorski E, Butigan T, Kolaric-Sviben G, Stevanović V, Gorenec L, Grgić I, Glavač F, Mehmedović A, Listeš E, Vilibić-Čavlek T. Severe West Nile Virus Neuroinvasive Disease: Clinical Characteristics, Short- and Long-Term Outcomes. Pathogens. 2022 Jan 2;11(1):52. doi: 10.3390/pathogens11010052. PMID: 35056000; PMCID: PMC8779330.
Zidovec-Lepej S, Vilibic-Cavlek T, Barbic L, Ilic M, Savic V, Tabain I, Ferenc T, Grgic I, Gorenec L, Bogdanic M, Stevanovic V, Sabadi D, Peric L, Potocnik-Hunjadi T, Dvorski E, Butigan T, Capak K, Listes E, Savini G. Antiviral Cytokine Response in Neuroinvasive and Non-Neuroinvasive West Nile Virus Infection. Viruses. 2021 Feb 22;13(2):342. doi: 10.3390/v13020342. PMID: 33671821; PMCID: PMC7927094.
Vilibic-Cavlek T, Zidovec-Lepej S, Ledina D, Knezevic S, Savic V, Tabain I, Ivic I, Slavuljica I, Bogdanic M, Grgic I, Gorenec L, Stevanovic V, Barbic L. Clinical, Virological, and Immunological Findings in Patients with Toscana Neuroinvasive Disease in Croatia: Report of Three Cases. Trop Med Infect Dis. 2020 Sep 14;5(3):144. doi: 10.3390/tropicalmed5030144. PMID: 32937866; PMCID: PMC7557803.
- We added a table that includes all available demographic and clinical characteristics of the analysed HCV patients (Table 1). We also included information about fibrosis stage reduction in the manuscript's main text (lines 155-158).
- We included the information about significant correlations between the analysed biomarkers and viremia levels (lines 191-195).
- We evaluated the ability of EGF and Ang-2 in predicting the fibrosis levels before antiviral treatment onset. This was done with binary logistic regression models using EGF and Ang-2 as sole predictors of the fibrosis stage. We calculated the sensitivity, specificity and AUC of the stated models and determined the best cutoff values for both EGF and Ang-2 (lines 275-282). We plotted the ROC curves for both models and included them in the manuscript's supplement (Figure S2, description added in the Supplementary Materials section, lines 424-427). We also described the utilized procedure for determining the best cutoff values in the Methods section (lines 140-141). Furthermore, we edited the abstract (line 26), discussion (line 404) and the conclusion (line 417) to emphasize the potential of EGF as a fibrosis biomarker in hepatitis C patients. We also created logistic regression models predicting fibrosis levels using other growth factors which were significantly different in patients with higher fibrosis stages (G-CSF, PDGF-AA and VEGF). However, these growth factors performed extremely poorly in separating the patients by fibrosis levels and were not included in the manuscript.
- All patients achieved sustained virological response (SVR) defined as undetectable HCV RNA at 12 weeks after therapy completion (line 85-87 in the revised version).
- Patients with known hematological, malignant (incl. hepatocellular carcinoma) or autoimmune diseases were excluded from the study (line 87-88 in the revised version).
- Considering that none of our patients exhibited an increase in fibrosis levels after antiviral therapy, resistance to therapy, or hepatocellular carcinoma, we were unable to analyse the predictive potential of cytokines and growth factors in the context of hepatitis C complications. However, we analysed the potential correlation between the measured cytokines and growth factors and the fibrosis level reduction after DAA treatment. The results of this analysis showed that Ang-2 measured before antiviral therapy was significantly higher in patients which exhibited reduced fibrosis levels after antiviral treatment (lines 260-263). This result suggested that Ang-2 measured before treatment could be used as a predictor for fibrosis reduction after antiviral therapy. However, the classification model utilizing Ang-2 for predicting fibrosis reduction after treatment performed poorly and was not included in the manuscript. Additionally, we would like to point out that the same cytokines and growth factors were significantly different in patients with higher fibrosis levels before and after antiviral treatment. Consequently, we only included the comparison between cytokine and growth factor levels before antiviral treatment.
Reviewer 2 Report
THere are my comments and suggestions
Line 91
To the title of paragraph 2.2, add serum
“Serum sample collection and virological monitoring”
Line 119-121
Put the sentences (Line 119-121)
“Liver fibrosis......[13]” in paragraph 2.1 Patients, and add at the end “according the METAVIR score[1].
Add the reference in bibliography section
1 Bedossa P, Poynard T. An algorithm for the grading of activity in chronic hepatitis C. The METAVIR Cooperative Study Group. Hepatology 1996; 24: 289-293 [PMID: 8690394 DOI: 10.1002/hep.510240201]
You can add the activity grade of the fibrosis if you know it.
Significant necrotic inflammation is defined by an activity grade (A1 to A3)
Line 139
Concerning the paragraph, 3.1, Patients' characteristics
In a Table, summarise all the patients' characteristics you know: number of males and females, their age, HCV genotype, METAVIR score, activity grade, alcohol consumption…
Line 151
To the title of the paragraph 3.2, add sera
Cytokine and growth factor levels in CHC patients’ sera before, during …..
Ligne 310 “Accordingly, a significant increase….CHC”
Please pay attention to the conclusion regarding the possible role of VEGF with its increased levels in patients with treatment (Figure 1). Compared to the control group (Figure 2), there is no significant increase. Edit the Sentence.
Ligne 320
To clarify, reformulate as following:
Ang-2 and SCF levels did not significantly change in response to DAA therapy.
Nevertheless, Ang-2 and SCF levels significantly increased compared to those of healthy individuals. These findings suggest that Ang-2 and SCF contribute to liver repair in CHC, regardless of DAA treatment.
Line 355
MMP-9 degrades extracellular matrix (ECM) proteins and activates cytokines and chemokines to regulate tissue remodeling.
Did you measure the expression of MMP-9 and TIMP-2?
These expression levels would provide additional information regarding the remodelling.
In any case, you should also discuss this point.
Lines 364, 365
“Finally, male patients…….female…”
Comment: Men generally consume more alcohol and pay less attention to their diet, which could also explain the more advanced fibrosis stages (F3-F4). It also depends on the duration of the HCV infection
So complete the phrase line 365 and write: “If not due to higher alcohol consumption and poor diet in the male patient group studied, this new finding should be evaluated in future research.”
Author Response
Reviewer 2
Line 91: We updated the title of paragraph 2.2. (line 94 in the revised version)
Line 119-121: We rephrased the sentence regarding liver fibrosis stages and moved the paragraph as suggested (lines 91-93 in the revised version). We also updated the reference as requested (reference number 13). Unfortunately, we did not measure the activity grade associated with fibrosis, which is why we did not include this information in the manuscript.
Line 139: We added a table that includes all available demographic and clinical characteristics of the analysed HCV patients (Table 1). We also included information about fibrosis stage reduction in the manuscript's main text (lines 155-158 of the revised version). Unfortunately, we did not record patients' alcohol consumption which is why we did not include this information in the manuscript.
Line 151: We updated the title of paragraph 3.2. (line 163 in the revised version).
Line 310: We apologise for not evaluating the recorded VEGF increase during antiviral treatment in the context of VEGF levels of the control group. We have edited the sentence and the conclusion associated with the finding (lines 337-341 of the revised version)
Line 320: We clarified the sentence as suggested (lines 350-356 of the revised version).
Line 355: Unfortunately, in this study, we did not measure expression of MMP-9 and TIMP-2, but in the future research we will conduct on chemokines, we will certainly include them.
Lines 364-365: We reformulated the sentence as suggested (lines 399-401 of the revised version).
Round 2
Reviewer 1 Report
The authors addressed most of my comments.